# The Role of Camellia Shell Substrates in Modulating the Nutritional Characteristics of *Pleurotus pulmonarius*

**DOI:** 10.3390/foods13182946

**Published:** 2024-09-18

**Authors:** Yikai Huang, Weike Wang, Na Lu, Jing Yu, Shaoning Chen, Zongsuo Liang

**Affiliations:** 1Zhejiang Province Key Laboratory of Plant Secondary Metabolism and Regulation, College of Life Sciences and Medicine, Zhejiang Sci-Tech University, Hangzhou 310018, China; huangyk1002@163.com; 2Hangzhou Academy of Agricultural Sciences, Hangzhou 310024, China; akeok@126.com (W.W.); 13738068366@163.com (N.L.); 3Key Laboratory for Quality Regulation of Tropical Horticultural Plants of Hainan Province, College of Horticulture, Hainan University, Haikou 570228, China; yujinghxy@163.com

**Keywords:** agricultural waste, substrates, edible fungi, agricultural traits, proximate composition

## Abstract

Camellia shells are the main by-product of camellia seed processing and are usually incinerated or disposed of as agricultural waste. In this study, camellia shells were employed in the *Pleurotus pulmonarius* cultivation process using five distinct formulae substituting for cottonseed shells. Our results show that as the substitution rate of camellia shells increased from 0% to 35%, the protein content in *P. pulmonarius* significantly increased from 34.05% to 53.35%. The polysaccharide content reached a peak value of 5.62% at 30% substitution of camellia shells. The DPPH free radical scavenging rate reached its maximum of 82.70% at 20% substitution of camellia shells. Furthermore, increases in the total amino acid contents in *P. pulmonarius* were positively correlated with the substitution rate of camellia shells. Considering the yield characteristics, the formula of 20% camellia shell substitution tested in this study appears to be optimal for *P. pulmonarius* cultivation. These findings not only provide a substrate to enhance the nutritional quality of *P. pulmonarius* but also demonstrate a novel approach for the ecological utilization of camellia shells.

## 1. Introduction

*Camellia oleifera* Abel, a genus of the Camellia family, is a major woody edible oil crop. This plant has been cultivated in China for more than 2000 years, which demonstrates its long-standing importance in the region [1]. As of 2023, the cultivation of *C. oleifera* in China spanned an area of approximately 4.6 million ha. This cultivation was mainly distributed in the Yangtze River Basin, with an oil production of close to 1 million tons [2]. Camellia shells are the main by-product of camellia seed processing, corresponding to approximately 0.54 tons of 1 ton of camellia fruit [3]. Most of these shells are either incinerated or discarded as agricultural waste, which leads to environmental pollution [4]. Therefore, searching for comprehensive methods for the utilization of camellia shells could potentially address these environmental concerns.

Camellia shells are nutrient-rich and contains cellulose, hemicellulose, lignin, tannin, pentose, ash, and a multitude of potassium carbonates. These nutrient components can adequately support the growth of mycelia and the development of fruiting bodies in the cultivation of edible fungi [5]. Given their high tannin content, camellia shells show an inhibitory effect on some contaminated bacteria in mushroom cultivation, which enhances their potential for the cultivation of different types of edible fungi, such as *Pholilta cylindracea*, *Hericium erinaceus*, *Lentinus edodes*, and *Pleurotus ostreatus* [6,7,8,9]. A previous study showed that when 20% camellia shells were used as a substrate for cottonseed shells, the resulting *Flammulina velutipes* fruit bodies exhibited increased nutritional content. Specifically, they contained higher levels of protein (18.16%), amino acids (13.14%), linoleic acid (0.49%), and linolenic acid (0.2%) compared to the control group [10]. Similarly, it was found that both treated and untreated camellia shell in the substrate resulted in accelerated growth of the mycelium and increased yield and nutrient contents of *P. geesteranus* (now named *P. pulmonarius*) using substrates with 78% proportions cottonseed shells combed with 20% rice bran [11]. However, the use of camellia shells in conjunction with various substrate formulations to cultivate *P. pulmonarius* remains uncommon.

*P. pulmonarius* is a type of an edible wood-decay fungus from the Basidiomycetes family that has been extensively cultivated in China since the 1990s [9]. It is highly valued for its protein-rich composition and pleasant umami flavor, which is attributed to its fiber and amino acid contents [12,13,14]. *P. pulmonarius* is known to be rich in multiple bioactive components. It has been significantly validated for its beneficial effects on liver protection, anti-cancer properties, and immune system enhancement [15,16]. The cultivation of *P. pulmonarius* utilizes raw materials such as wood chips and cottonseed shells as substrates, as per current practices [17,18,19]. However, as the scale of cultivation continues to expand, the availability and seasonality of raw materials have led to a yearly increase in their prices, which has escalated production costs. Therefore, to ensure the sustainable development of the *P. pulmonarius* industry and enhance nutrient composition, it may be both important and feasible to explore the use of local agricultural waste as substrates.

In this study, we utilized camellia shells as a medium to cultivate *P. pulmonarius*, employing five distinct formulae with varying substitution rates of cottonseed shells. We conducted an analysis of the nutrient composition, hydrolyzed amino acid assay, and antioxidant capacity to assess the nutritional characteristics of *P. pulmonarius*. This study not only provides a substrate to improve the nutritional quality of *P. pulmonarius* but also presents a novel approach for the utilization of camellia shells. This approach strikes a balance between generating economic benefits and maintaining ecological sustainability.

## 2. Materials and Methods

### 2.1. Preparation of P. pulmonarius Strain and Growing Substrates

The *P. pulmonarius* strain (Hangxiu 1) was generously provided by the Hangzhou Academy of Agricultural Sciences, located in Hangzhou, China. This strain was routinely cultivated on potato dextrose agar slants. The substrate was supplemented with agricultural waste camellia shells, which were purchased from Quzhou in Zhejiang province.

We designed formulations with different ratios of camellia shell to replace cottonseed shell on traditional formulations used for *P. pulmonarius* cultivation (Table 1). The water content in the substrate was maintained between 60% and 65%, and the pH was controlled within the range of 6.0 to 7.0. Approximately 1.3 to 1.4 kg of the moistened substrates were packed into polypropylene bags measuring 17 × 38 cm. To eliminate bacterial contamination and maintain the nutrients inside the bag, these bags were then sterilized at a temperature of 126 °C for 2 h. To facilitate spawn inoculation and provide aeration, a single vertical hole was made in the center of each bag.

### 2.2. Cultivation of P. pulmonarius

The substrate was inoculated with grain spawn, and bags were incubated for 25 days at a temperature of 25 °C and a relative humidity of 60%–70%. The mycelial growth rate was recorded on a weekly basis until the substrate was fully colonized with mycelia. Then, the bags were cooled for 3 days, and the plugs were removed when primordial bodies emerged. The Fruit bodies of *P. pulmonarius* were harvested during the first flush. The yield of fresh mushrooms was calculated based on the weight per bag. Biological efficiency (BE) was determined as the ratio of the fresh mushroom weight to the dry weight of the substrate.

### 2.3. Determination of Proximate Composition

The nutrient contents were determined in accordance with the National Standards of the People’s Republic of China. The moisture content was measured as per national standard GB5009.3, and the aqueous extract content was measured as previously described by GB/T 8305 [20,21]. The ash and crude fiber contents were determined according to national standards GB 5009.4 and GB 5009.10, respectively [22,23]. Crude polysaccharide was determined using phenol-sulfuric acid colorimetry according to national standard SN/T 4260 [24]. The protein content was determined using the Kjeldahl nitrogen determination method with a conversion factor of 6.25 according to the method described by AOAC [25].

### 2.4. Hydrolyzed Amino Acid Assay

The hydrolyzed amino acid content of the mushrooms was determined according to national standard GB 5009.124 [26]. Finely milled dry mushroom powders (5.0 g) were extracted with 10 mL of 6 mol L^−1^ hydrochloric acid and shaken for 22 h at a temperature of 110 °C. The extract was then vacuum-dried at 50 °C using an evaporator, then dissolved in 2.0 mL sodium citrate buffer (pH 2.2). The extracted solutions were filtered through a 0.22 μm hydrophilic membrane and analyzed by an automatic amino acid analyzer (L-8900; Hitachi Ltd., Chiyoda, Japan).

### 2.5. Antioxidant Capacity

The mushroom samples (0.2 g) were subjected to ultrasound extraction for 30 min, then filtered and adjusted to a volume of 10 mL. The antioxidant activity of these aqueous extracts was evaluated using both ABTS and 2, 2-diphenyl-1-picrylhydrazyl (DPPH) radicals methods.

To determine the DPPH free radical scavenging rate, 2 mL of the mushroom aqueous extracts from each group were mixed with DPPH solution, then incubated in the dark at room temperature for 30 min. The scavenging activity of DPPH radicals was measured using the absorbance value at 517 nm. DPPH free radical scavenging rates were calculated according to the following formula:(1)$$Scavenging\;Rate=\frac{Absorbance\;of\;liquid\;to\;be\;tested-control\;group}{absorbance\;of\;the\;blank\;group}$$

To determine the ABTS free radical scavenging rate, 2.9 mL ABTS solution was mixed with 0.1 mL mushroom aqueous extracts. After reacting at room temperature for 30 min, the absorbance was measured at 734 nm. Distilled water was used as the control treatment.

### 2.6. Statistical Analysis

All data were statistically analyzed using SPSS 26 software (IBM, New York City, NY, USA). The data were analyzed using a one-way analysis of variance (ANOVA) and are expressed as the means ± standard deviations of three replicates. The mean differences between individual groups were separated using Fisher’s Least Significant Difference (LSD) test at a confidence level of 95%.

## 3. Results and Discussion

### 3.1. Effect of Camellia Shell Substrate on Yield and Biological Efficiency (BE)

To assess the effect of camellia shell substrate on mushroom growth, mycelial growth rates were recorded and calculated. The control formulae exhibited a mycelial growth rate of 0.77 mm/d, which was significantly higher than that of other treatments (Table 2). When using a non-supplemented camellia shell substrate (Treatment A1), the yield of *P. pulmonarius* (415.10 g/bag) did not show a significant difference compared to that of Treatment A2 (412.20 g/bag) and Treatment A3 (407.83 g/bag). The biological efficiency was highest in the control group (74.13%), followed by treatment A2 (73.57%) and treatment A3 (72.83%). No significant difference (*p* < 0.05) was observed between the biological efficiency of *P. pulmonarius* cultivated on the control substrate and those on substrates A2 and A3. The results show that the addition of a small quantity of camellia shell to the substrate (ranging from 0 to 20%) did not significantly impact the yield and biological efficiency of artificially cultivated *P. pulmonarius* (Table 2).

Previous studies have shown that the inclusion of camellia shells in a certain proportion may inhibit mushroom growth and decrease the yield and biological efficiency (BE) of fruiting bodies. For example, when the addition of camellia shell reached 78% of the substrate, the mycelial growth rate *of Lentinula edodes* was recorded to be only 4.75 mm/day [27]. Similarly, the yield of *Hericium erinaceus* was found to be inversely proportional to the quantity of camellia shells added to the formula, with the growth of the fruiting body inhibited when the proportion of camellia shells reached 43% [16,28]. In this study, the addition of camellia shells resulted in a yield of *P. pulmonarius* ranging from 163.9 g/bag to 412.2 g/bag compared to a yield of 415 g/bag in the control group. When the substitution ratio of camellia shells was less than 30%, there was no significant decrease in the yield relative to the control group. The BE of *P. pulmonarius* varied from 73.57% to 29.27% when camellia shells were added to the substrates, which is lower than that of the control group (74.13%), although not corresponding to a significant decrease. BE is an important indicator in evaluating the economic benefits and sustainable development of edible fungus production. Factors influencing BE include medium composition, cultivation conditions, strain selection, and inoculation volume. In a culture medium formulated with conventional materials such as sawdust and wheat straw, a BE value greater than 50% is considered to be profitable [29]. The BE of *P. pulmonarius* grown on substrates with added pine, poplar, and honeysuckle rattan varied from 61.89% to 81.01% [30]. This study found that the BE of the A2, A3, and A4 groups were in a reasonable range. Although the yield and BE decreased with increases in the ratio of camellia shell supplementation, there was no significant difference between the control and A2 and A3 groups, indicating that camellia shells may be used as a potential substrate material for *P. pulmonarius* cultivation.

### 3.2. Camellia Shell Substrates Affect Moisture Content and Aqueous Extract Content

The moisture content and aqueous extract content of *P. pulmonarius* were determined, as shown in Table 3. The moisture content of the fruiting bodies was observed to be the highest in treatment A3 (89.26%), followed by A2 (89.02%) and A4 (88.95%). The moisture contents of A5 and A6 were recorded as 87.73% and 86.69%, respectively, indicating no significant increase compared to the control group (A1). The aqueous extract contents of A2 and A3 were found to be 7.22% and 5.51%, respectively, which are significantly lower (*p* < 0.05) than that of the control group. This indicates that the addition of camellia shell to substrates resulted in a decrease in the aqueous extract content of *P. pulmonarius*.

### 3.3. Camellia Shell Substrates Improve the Contents of Proximate Composition

Camellia shells are rich in cellulose, hemicellulose, and lignin, which makes them similar to the cultivation-medium ingredients for edible fungi [31]. They also contain large amounts of bioactive substances such as tannins, saponins, flavonoids, and polysaccharides, which can enhance the nutrient quality of edible fungi [32]. The composition of the substrate can cause significant differences in the contents of these bioactive ingredients [33,34]. To assess the nutrient contents in different groups, the ash, crude fiber, crude polysaccharide, and protein contents of *P. pulmonarius* were evaluated (Figure 1). The ash content in the control group was recorded as 5.58%, which is significantly lower (*p* < 0.05) than the other treatments. The ash content of *P. pulmonarius* with the addition of camellia shells to the substrate ranged from 6% to 7%. These results indicate that the *P. pulmonarius* grown on the substrate with camellia shells contained more inorganic components than that of fungi grown on the control substrate [35]. The ash contents of *P. pulmonarius* grown on substrates based on sawdust, cottonseed shells, and brans ranged from 6.43% to 6.56%, which is slightly less than that of the fungi grown on camellia shells [36]. The highest crude fiber content was observed in treatment A6 (8.84%), followed by treatment A3 (8.65%) and treatment A2 (8.57%), which were significantly higher than that of the control group. The crude polysaccharide content of *P. pulmonarius* reached its highest peak (5.62%) when the camellia shell substitution rate was 20% (Treatment A3). The crude polysaccharide contents of treatments A2 and A3 were significantly higher (*p* < 0.05) than that of the control group. However, when the substitution rates of camellia shells exceeded 35%, the crude polysaccharide contents of *P. pulmonarius* were significantly lower (*p* < 0.05) than those of the control group. The protein contents of *P. pulmonarius* increased significantly (*p* < 0.05) with the increase in the camellia shell substitution rate from 0% to 35%, and the content reached a maximum of 53.35% at a 35% substitution rate.

The protein content in edible mushrooms ranges from 19–35%, with a digestibility rate of 72–83%. This makes edible mushrooms an excellent source of protein [37]. On a maize straw substrate, *Pleurotus florida* had a protein content of 48.79%, while on a spadix substrate, it had a protein content of 23.41% [38]. A wheat bran-supplemented substrate demonstrated a high value of protein content of 19.14% [17]. In this study, protein contents from *P. pulmonarius* fruit bodies were assessed using Kjeldahl nitrogen determination. The results indicate that *P. pulmonarius* fruit bodies grown on a 35% camellia shell substrate had the highest protein content of about 53.35—29.2% higher than that of the control group.

Fungal polysaccharides are recognized as valuable medicinal compounds and are often referred to as “biological response modifiers” due to their pharmacological functions. These functions include anti-aging, anti-tumor, and anti-inflammation properties [15,39,40,41,42]. The supplementation of camellia shells has been found to enhance the biosynthesis of these polysaccharides in *P. pulmonarius*. Furthermore, polysaccharides from *P. pulmonarius* (PFP) have also demonstrated their potential biological activities, such as antioxidant, immunomodulatory, hypolipidemia, and hepatoprotection properties, which have attracted great interest in the nutrition and health industries [43]. It was reported that *P. pulmonarius* had 2.89% polysaccharide content when using asparagus straw as a raw material. *P. pulmonarius* cultivated on mulberry twigs contained 3.02% polysaccharides [44]. In this study, we observed that the crude polysaccharide content in *P. pulmonarius* fruit body reached a peak value of 5.62% when cultivated on a substrate with 30% substitution of camellia shells. This indicates that 30% camellia shell supplementation significantly improved the polysaccharide contents in *P. pulmonarius*. Considering that the yield of A3 showed no significant decrease compared to the control, it can be inferred that formula A3 could serve as an effective cultivation substrate to improve polysaccharide contents.

### 3.4. Camellia Shell Substrates Enhance the Antioxidant Activities of Mushroom Aqueous Extracts

The antioxidant activities of fruit bodies in substrates A2, A4, and A5 exhibited a significant increase compared to that of the control group in both DPPH and ABTS assays (Table 4). The DPPH free radical scavenging rate of the aqueous extract of *P. pulmonarius* in treatments A2, A3, A4, and A5 increased significantly (*p* < 0.05) compared to the control group and reached the peak of 82.7% when the camellia shell substitution rate was 20% (Treatment A3).

In recent years, the demand for antioxidants has increased due to their potential in commercial applications and their health-promoting properties [45,46]. Various types of mushrooms have demonstrated their antioxidant properties, with protective capacity against oxidation stress. Therefore, they have received considerable attention as a commercial source of antioxidants in the neutralization of free radicals in the body [47,48,49,50,51]. The effects of different substrates on the antioxidant properties of mushrooms might be very significant. The antioxidant activity of *P. ostreatus* grown on a substrate mixture with 80% wheat straw and 20% peat moss was 7.8 mg mL^−1^, while that of *P. ostreatus* grown on a substrate mixture with 60% wheat straw and 40% peat moss was 7.10 mg mL^−1^ [52]. Three different types of contaminant waste, namely pine sawdust, coconut coir, and waste paper, were used as the substrate to cultivate *P. ostreatus*. The coconut-coir substrate presented the highest antioxidant activity, followed by the pine sawdust and, finally, the waste paper [53]. The present study evaluated the antioxidant properties of *P. pulmonarius* fruit bodies grown on substrates with different formulae using DPPH and ABTS free radical scavenging assays. Treatment A3 (20% camellia shell supplement) presented the highest inhibition for DPPH, at 82.70%, and A6 presented the highest inhibition for ABTS, at 54.89%. These results indicate that camellia shell supplement may promote the antioxidant properties of *P. pulmonarius*; considering the yield and chemical composition of *P. pulmonarius* grown on different formulae the A3 formula might be the optimal substrate.

### 3.5. Camellia Shell Substitution Affects the Amino Acid Content in Fruit Bodies of P. pulmonarius

The total amino acid (TAA) and contents of 16 amino acids in *P. pulmonarius* were determined by an amino acid analyzer (Table 5). The TAA content of *P. pulmonarius* exceeded that of the control group when the camellia shell substitution rate was 20% or more. The highest content was observed in treatment A6 (24.83%), followed by treatments A5 (23.60%) and A4 (23.00%). The TAA content of *P. pulmonarius* showed a positive correlation when the substitution rate of added camellia shells was 20% or more. The essential amino acid (EAA) content of the fruit bodies grown on the A3 substrate was 1.1 fold of that of the control group and reached its highest value on the A6 substrate, at about 8.7% (Figure 2). Among the 16 amino acids, HIS content was found to be highest in the A4 (0.70%) and A6 (0.74%) substrates. The contents of the other 15 amino acids in *P. pulmonarius* were higher than those in the control group when the camellia shell substitution rate was 20% or more and positively correlated with the substitution rate of added camellia shells. Both the total amino acid contents and the contents of the 16 amino acids were highest when the camellia shell substitution rate was 39% (A6).

Previous studies demonstrated that substrate components may also affect the amino acid composition [54]. Cultivation of *P. pulmonarius* with water hyacinth soaked in biogas fluid results in an increase of 23–35% in total amino acids over its counterpart using sawdust alone as the substrate [17]. *P. pulmaonarius* mushrooms grown on pine waste showed a high total amino acid content of 29.71%—8.88% higher than that of the control group [30]. In the present study, we found that, when grown on substrates supplemented with camellia shells, *P. pulmonarius* had a total amino acid content of 24.9%, which is 5.9% higher than that of the control group. Our result suggests that adding 35% camellia shells to the substrate produces mushrooms of superior protein quality.

Above all, this innovative substrate could offer improved nutrient profiles, better antioxidant activities, and higher amino acid contents, all of which are critical factors in the nutritional assessment of high-quality mushrooms. Furthermore, the utilization of camellia shell substrates may lead to more sustainable agricultural practices by reducing waste and promoting the circular economy within the agricultural sector. Further studies may uncover additional benefits, such as increased yield, enhanced mushroom flavor, and even medicinal properties, expanding the horizons of fungal cultivation and contributing to both environmental and economic sustainability.

## 4. Conclusions

In this study, we conducted a comprehensive evaluation of the impacts of substituting camellia shells in substrates on the chemical and nutritional characteristics of *P. pulmonarius*. This was achieved by investigating several factors. such as yields, proximate composition, antioxidant properties, and amino acid contents. The results showed significant improvements in the protein, polysaccharide, and total amino acid contents when camellia shell substrates were used. Taking into account both yield and nutritional characteristics, our study found that a 20% substitution of camellia shell in the substrate is optimal for the cultivation of *P. pulmonarius*. This resulted in a yield of 407.83 g/bag and a Biological Efficiency (BE) of 72.83%, along with the highest crude polysaccharide content and DPPH inhibition. In conclusion, our findings suggest that camellia shell substrates are not only effective but also economical for the cultivation of *P. pulmonarius*. Moreover, this study provides valuable insights into the potential environmental applications of agricultural camellia shells.

## Figures and Tables

**Figure 1 foods-13-02946-f001:**
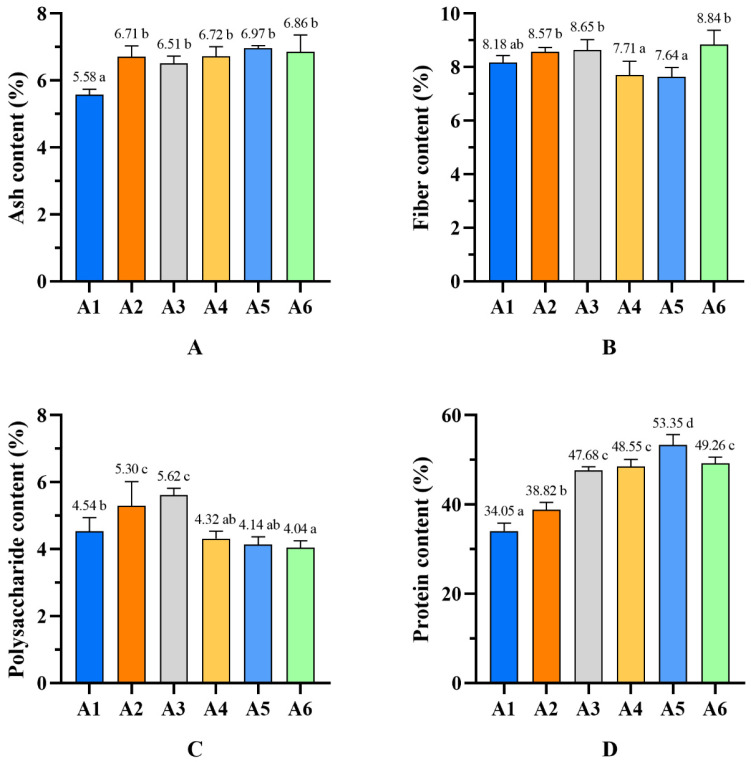
The contents of ash (**A**), fiber (**B**), polysaccharides (**C**), and protein (**D**) in *P. pulmonarius*. Different letters represent significant differences at the *p* < 0.05 level.

**Figure 2 foods-13-02946-f002:**
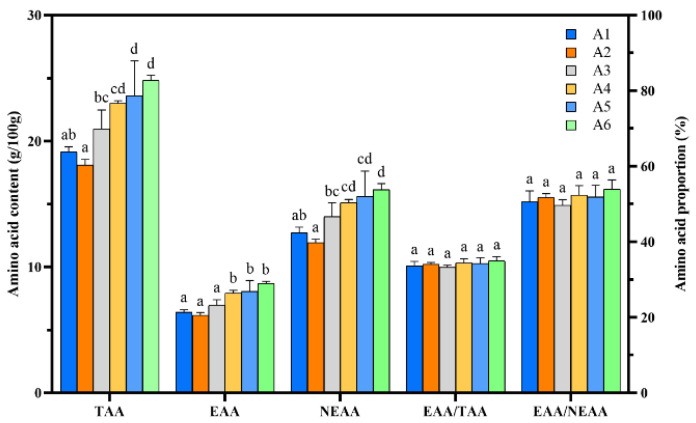
The amino acid content and amino acid proportion of *P. pulmonarius*. TAA, total amino acids; EAA, essential amino acid; NEAA, non-essential amino acid. Different letters represent significant differences at the *p* < 0.05 level.

**Table 1 foods-13-02946-t001:** The contents of components in each substrate.

Treatment	Content (g/100 g Dry Weight)					
	Camellia Shell	Cottonseed Shell	Sawdust	Bran	Sucrose	Gypsum
A1	0	39	39	20	1	1
A2	10	29	39	20	1	1
A3	20	19	39	20	1	1
A4	30	9	39	20	1	1
A5	35	4	39	20	1	1
A6	39	0	39	20	1	1

**Table 2 foods-13-02946-t002:** The mycelium running, yield, and biological efficiency of *P. pulmonarius*.

Sample	Mycelium Running (cm/d)	Total Yield (g/bag)	Biological Efficiency (%)
A1	0.54 ± 0.07 d	415.10 ± 40.63 b	74.13 ± 7.26 c
A2	0.36 ± 0.04 c	412.20 ± 39.35 b	73.57 ± 7.02 c
A3	0.35 ± 0.06 c	407.83 ± 24.93 b	72.83 ± 4.45 c
A4	0.33 ± 0.24 bc	307.10 ± 53.87 b	54.84 ± 9.62 b
A5	0.29 ± 0.20 ab	246.07 ± 19.59 b	43.94 ± 3.50 b
A6	0.27 ± 0.24 a	163.90 ± 21.51 a	29.27 ± 3.84 a

Different letters represent significant differences at the *p* < 0.05 level.

**Table 3 foods-13-02946-t003:** The moisture content and aqueous extract content of *P. pulmonarius*.

Sample	Moisture Content (%)	Aqueous Extract Content (%)
A1	87.05 ± 0.95 a	50.55 ± 1.24 b
A2	89.02 ± 0.48 b	43.33 ± 0.57 a
A3	89.26 ± 0.55 b	45.04 ± 1.43 a
A4	88.95 ± 0.54 b	43.69 ± 1.05 a
A5	87.73 ± 0.85 a	43.10 ± 3.11 a
A6	86.69 ± 1.36 a	45.61 ± 1.58 a

Different letters represent significant differences at the *p* < 0.05 level.

**Table 4 foods-13-02946-t004:** The oxidation resistance of *P. pulmonarius*.

Sample	DPPH Free Radical Scavenging Rate (%)	ABTS Free Radical Scavenging Rate (%)
A1	61.77 ± 6.02 a	40.41 ± 5.88 b
A2	73.90 ± 4.40 bc	47.27 ± 1.79 cd
A3	82.70 ± 2.02 d	30.96 ± 5.34 a
A4	76.30 ± 1.71 c	45.97 ± 2.61 c
A5	77.63 ± 2.83 cd	51.71 ± 4.59 de
A6	67.81 ± 7.70 ab	54.89 ± 6.95 e

Different letters represent significant differences at the *p* < 0.05 level.

**Table 5 foods-13-02946-t005:** The amino acid contents of *P. pulmonarius*.

Component	Content (g/100 g Dry Weight)					
	A1	A2	A3	A4	A5	A6
Asp	2.02 ± 0.09 ab	1.75 ± 0.05 a	2.09 ± 0.18 b	2.22 ± 0.05 bc	2.23 ± 0.22 bc	2.43 ± 0.10 c
Thr	1.04 ± 0.02 ab	0.94 ± 0.02 a	1.10 ± 0.07 bc	1.20 ± 0.01 cd	1.22 ± 0.11 cd	1.31 ± 0.03 d
Ser	1.05 ± 0.03 ab	0.92 ± 0.03 a	1.09 ± 0.06 b	1.17 ± 0.04 bc	1.17 ± 0.11 bc	1.27 ± 0.07 c
Glu	4.77 ± 0.45 a	4.75 ± 0.13 a	5.65 ± 0.32 ab	5.96 ± 0.39 b	6.41 ± 0.85 b	6.04 ± 0.59 b
Gly	0.97 ± 0.02 a	0.99 ± 0.02 ab	1.11 ± 0.06 bc	1.22 ± 0.03 cd	1.25 ± 0.12 d	1.31 ± 0.01 d
Ala	1.14 ± 0.04 a	1.13 ± 0.03 a	1.25 ± 0.05 b	1.36 ± 0.01 c	1.40 ± 0.08 c	1.51 ± 0.05 d
Val	1.08 ± 0.00 ab	1.03 ± 0.02 a	1.18 ± 0.05 b	1.32 ± 0.00 c	1.34 ± 0.11 cd	1.43 ± 0.03 d
Met	0.21 ± 0.02 a	0.20 ± 0.00 a	0.24 ± 0.03 ab	0.25 ± 0.02 ab	0.26 ± 0.06 ab	0.29 ± 0.03 b
Ile	0.83 ± 0.01 ab	0.81 ± 0.02 a	0.92 ± 0.04 b	1.04 ± 0.01 c	1.06 ± 0.09 c	1.15 ± 0.02 d
Leu	1.32 ± 0.04 a	1.34 ± 0.04 a	1.45 ± 0.07 a	1.67 ± 0.06 b	1.68 ± 0.11 bc	1.82 ± 0.02 c
Tyr	0.47 ± 0.01 a	0.51 ± 0.02 ab	0.55 ± 0.04 ab	0.60 ± 0.05 bc	0.68 ± 0.08 c	0.70 ± 0.06 c
Phe	0.87 ± 0.03 a	0.85 ± 0.02 a	0.94 ± 0.05 a	1.07 ± 0.05 b	1.11 ± 0.10 b	1.17 ± 0.06 b
Lys	1.07 ± 0.06 a	1.00 ± 0.05 a	1.13 ± 0.07 a	1.36 ± 0.07 b	1.40 ± 0.15 b	1.52 ± 0.12 b
His	0.68 ± 0.09 ab	0.53 ± 0.08 a	0.58 ± 0.05 ab	0.70 ± 0.08 ab	0.64 ± 0.07 ab	0.74 ± 0.09 b
Arg	0.86 ± 0.06 a	0.68 ± 0.03 ab	0.90 ± 0.12 bc	0.95 ± 0.04 bc	0.93 ± 0.15 bc	1.11 ± 0.07 c
Pro	0.74 ± 0.05 a	0.68 ± 0.02 a	0.79 ± 0.06 ab	0.91 ± 0.04 c	0.90 ± 0.07 bc	1.00 ± 0.05 c
Total	19.13 ± 0.34 ab	18.10 ± 0.36 a	20.97 ± 1.22 bc	23.00 ± 0.16 cd	23.60 ± 2.26 d	24.83 ± 0.33 d

Different letters represent significant differences at the *p* < 0.05 level.

## Data Availability

The original contributions presented in the study are included in the article, further inquiries can be directed to the corresponding author.

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
