# Peer review of "The Role of Camellia Shell Substrates in Modulating the Nutritional Characteristics of Pleurotus pulmonarius"

_foods, 2024, doi:10.3390/foods13182946_

Round 1
Reviewer 1 Report
Comments and Suggestions for Authors
Specific comments:
Abstract section
It is necessary to improve this section, by reducing the information presented, the text is too long
Introduction section
Lines 54-55: it is uncommon? or inexistent??.... if there are investigations about it it is important to mention this
Lines 46-47: mention the bacteria
Camelia shells are used for other food products???
Material and methods
Lines 85-86: the sterilization does not compromise the nutritional composition of the bag content.? Besides, why this step is important for the experiment, please explain
Results and discussion
Lunes 154-156: please add more details about this statement
Lines 173-181: please add the discussion of this part, it is only a description of the results
Author Response
We greatly appreciated the reviewers’ comments and kind suggestions on our manuscript “The Role of Camellia Shell Substrates in Modulating the Nutritional Characteristics of Pleurotus pulmonarius”. These comments are all very helpful for improving our paper, and for providing important guidance to our researches. We carefully revised the manuscript according to all your comments and suggestions, and improved modifications have been considered in this resubmitted version. The original reviewers’ comments are in italics. Responses are shown in blue.
The following is the point to point response to the comments of reviewer:
Comment 1: It is necessary to improve this section, by reducing the information presented, the text is too long
Response 1: Thanks very much for the kind comments. The abstract section of the article has been rewritten and condensed. (Page 1 Line 12~23)
Comment 2: Lines 54-55: it is uncommon? or inexistent??.... if there are investigations about it it is important to mention this.
Response 2: Thanks for comments. The utilization of the camellia shells in the cultivation of Pleurotus pulmonarius is still uncommon. Camellia shells has been used to cultivate different types of mushrooms, such as Pholilta cylindracea, Hericium erinaceus, Lentinus edodes, Pleurotus ostreatus and Pleurotus geesteranus. Previously, the treated or untreated camellia shells were used to substituted cottonseed hull to cultivate P. geesteranus (now named Pleurotus pulmonarius), which has the different materials and design from our study (Zhang, 2019). We mentioned it and revised the sentence as following: “Similarly, it was found that the treated or untreated camellia shell in substrate showed a accelerated growth of the mycelial and increase yield, nutrients contents of P. geesteranus (now named P. pulmonarius), using substrates with 78% proportions cottonseed hull combining 20% rice bran. However, the use of camellia shells in conjunction with various substrate formulations to cultivate P. pulmonarius remains uncommon.” (Page 2 Line 45~49)
Comment 3: Lines 46-47: mention the bacteria
Camelia shells are used for other food products???
Response 3: Thanks so much for comments. Camelia shells are always used in many industries such as making tannins, furfural, activated carbon, xylooligosaccharides, and extracting saponin. For food production, camelia shells are mainly used as an additive of edible fungi substrate. Since camelia shell contain saponin, it is hard to be directly used in food production. To express more accurately, we revised the sentence as following: “The camellia shells with their high tannin content show an inhibitory effect on some contaminated bacteria in mushrooms cultivation which enhances their potential for cultivating different types of edible fungi, such as Pholilta cylindracea, Hericium erinaceus, Lentinus edodes, and Pleurotus ostreatus.” (Page 2 Line 38~41)
Comment 4: Material and methods
Lines 85-86: the sterilization does not compromise the nutritional composition of the bag content.? Besides, why this step is important for the experiment, please explain
Response 4: Thanks for comments. Sorry for my ambiguous expression. In fact, the process of sterilization is meticulously designed to eliminate harmful pathogens and microorganisms without degrading the essential vitamins, minerals, and other nutrients inside the bag. The high temperatures used in sterilization are carefully controlled to ensure that they are sufficient to kill bacteria and other contaminants, yet not so extreme as to break down the delicate molecular structures of the nutritional composition. We have revised this sentence as following: “ To eliminate bacterial contamination and maintain the nutrients inside the bag, these bags were then sterilized at a temperature of 126 ℃ for 2 hours.”(Page 4 Line 78~79)
Comment 5: Lunes 154-156: please add more details about this statement
Response 2: Thanks very much for comments. Details of the discussion of biological efficiency have been added as following: “The BE is an important indicator for evaluating the economic benefits and sustainable development of edible fungi production. Factors influencing the BE include medium composition, cultivation conditions, strain selection, and inoculation volume. In a culture medium formulated with conventional materials such as sawdust and wheat straw, the BE value greater than 50% is considered to be profitable. The BE of P. pulmonarius grown on the substrates added pine, poplar, and honeysuckle rattan varied from 61.89% to 81.01%”. (Page 7 Line 150~156)
Comment 6: Lines 173-181: please add the discussion of this part, it is only a description of the results
Response6: Thanks very much for the kind comments. Details of the discussion of the contents in Pleurotus pulmonarius have been added as: “This result indicated that the P. pulmonarius grown on the substrate with camellia shells contained more inorganic components than that of fungi on the control substrate. The ash content of P. pulmonarius grown on the substrate based on the sawdust, cottonseed shells and brans range from 6.43% to 6.56%, lightly less than that of the one grown on the camellia shells”. (Page 8 Line 181~185)

Reviewer 2 Report
Comments and Suggestions for Authors
The manuscript "The Role of Camellia Shell Substrates in Modulating the Nutritional Characteristics of Pleurotus pulmonarius" is very interesting to me because it finds the added value of waste material, which is suitable for organic agriculture.
Title: correct Pleurotus Pulmonarius to Pleurotus pulmonarius
Rename 2.1. Preparation of Microorganism strain and Substrates to
2.1. Preparation of P. pulmonarius strain and growing substrates
In my opinion, the figures are not adequate for this manuscript. It is better to present the results in a table.
Before the conclusion, add a paragraph about the perspectives and possibilities of using camellia shell substrates in the production of Pleurotus pulmonarius. Real potential. Apostrophize the significance of the study!
Author Response
We greatly appreciated the reviewers’ comments and kind suggestions on our manuscript “The Role of Camellia Shell Substrates in Modulating the Nutritional Characteristics of Pleurotus pulmonarius”. These comments are all very helpful for improving our paper, and for providing important guidance to our researches. We carefully revised the manuscript according to all your comments and suggestions, and improved modifications have been considered in this resubmitted version. The original reviewers’ comments are in italics. Responses are shown in blue.
The following is the point to point response to the comments of reviewer:
Comment 1: Title: correct Pleurotus Pulmonarius to Pleurotus pulmonarius
Response 1: Thanks for your comments! This mistake has been corrected. (Page 1 Line 2)
Comment 2: Rename 2.1. Preparation of Microorganism strain and Substrates to
2.1. Preparation of P. pulmonarius strain and growing substrates
Response 2: Thanks. This has been revised. (Page 3 Line 69)
Comment 3: In my opinion, the figures are not adequate for this manuscript. It is better to present the results in a table.
Response 3: Thanks for your comments. According to your suggestion, we converted the Fig.1 and Fig. 3 in tables. For Fig. 2, we added the detail data on the top of bar chart to show the differences between different treatments intuitively. (Page 8 Table 3; Page 11 Table 4; Page 10 Fig. 1)
Comment 4: Before the conclusion, add a paragraph about the perspectives and possibilities of using camellia shell substrates in the production of Pleurotus pulmonarius. Real potential.
Response 4: Thanks for your comments! It’s very helpful for improving our paper. This part has been added. (Page 13~14 Line 276~282)
